# Breaking the Cycle: A Novel Approach to Nash Equilibria in Both Cyclical and Non-Cyclical Noncooperative Games

## Abstract

This paper presents a novel perspective on finding Nash Equilibria (NE) in non-cooperative games, arguing that cyclical strategies are not chaotic anomalies but orderly sequences integral to an equilibrium. We establish the theoretical equivalency between a complete set of cyclical strategies and the support set of a Mixed Strategy NE (MSNE). Our proof demonstrates that cyclical strategies must form a circular counter, implying that a complete set is necessary to support a MSNE due to the intrinsic counterbalancing dynamic. This insight enables a novel graph search learning representation of self-play that finds an NE as a graph search. We empirically validate our approach by demonstrating improved self-play efficiency in discovering both a Pure Strategy NE (PSNE) and a MSNE in noncooperative games such as Connect4 and Naruto Mobile. Our findings offer a more efficient alternative to existing methods like Policy Space Response Oracles (PSRO), which can be computationally demanding due to the expanding population of opponents. This research contributes to the fundamental challenge of finding NE in game theory and artificial intelligence, offering a more efficient and theoretically grounded approach.

## 1 Introduction

In 1951, Nash proved that every finite noncooperative game has at least one equilibrium point, known as Nash Equilibrium (NE) (Nash, 1951). A NE is a situation where no player can benefit from changing their strategy unilaterally. This makes NE a desirable solution concept for various fields, such as game theory, economics, and artificial intelligence. NE can help resolve conflicts among rational agents by ensuring that no one has an incentive to deviate from their agreed strategy. However, finding the NE for a given game is not trivial. In some complex games, such as Go and Poker, the optimal strategy may involve randomization and uncertainty. In these cases, we may need to use self-play to simulate and discover the NE.

There are two main approaches to self-play: Policy Space Response Oracles (PSRO) (Lanctot et al., 2017) and AlphaZero (Silver et al., 2017). PSRO is a general framework that can find both deterministic Pure Strategy Nash Equilibrium (PSNE) and probabilistic Mixed Strategy Nash Equilibrium (MSNE). AlphaZero is a more efficient method, but it only works for games with PSNE.

As shown in Figure 1, PSRO finds a NE by increasing the number of opponents in self-play. The Double Oracle (DO) method (McAleer et al., 2021) ensures that the self-play does not encounter cycles, where policies keep countering each other endlessly. Instead, by learning the best response against all policies in the self-play sequence, the self-play converges to a NE.

However, as the number of policies grows, the agents face a larger and larger pool of opponents. For example, in the OpenAIFive project (OpenAI et al., 2019), the agents had to deal with up to 14,000 opponent policies. This means that to find the next best response, an agent has to sample and calculate the best response over all 14,000 policies. This is computationally expensive and time-consuming, especially when the sampling has to be repeated many times to adjust the distribution of the policies. Therefore, there is a need for more efficient methods to handle cyclical and probabilistic strategies.

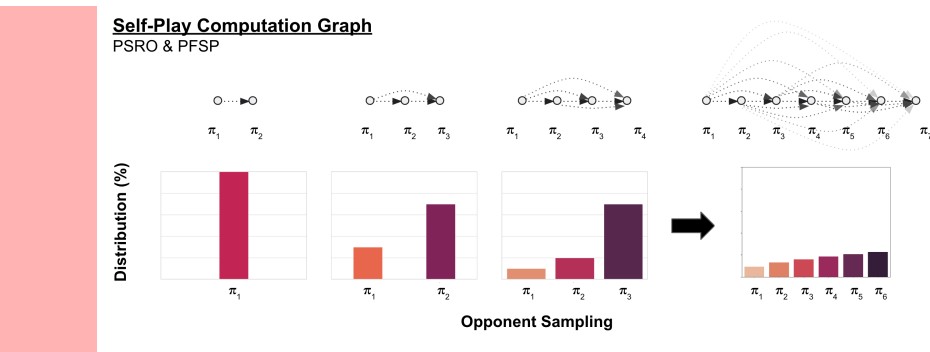

Figure 1: An illustration of PSRO - PSRO finds a NE by letting agents learn better policies through self-play. In each round, an agent improves its current policy by learning the best response against a probabilistic sample of all previous policies in the self-play sequence. The new policy is then added to the set of previous policies for the next round.

In contrast, the AlphaZero project (Silver et al., 2017) has shown that in games where randomness does not influence the outcome, it is possible to develop a self-play algorithm that maintains a consistent compute complexity.

AlphaZero is particularly effective in deterministic games like Go, Chess, and Shogi, where the gameplay is not affected by randomness. In these games, all NE are deterministic PSNE, which means players can make optimal decisions based on perfect information. AlphaZero incorporates this determinism into its self-play mechanism, resulting in a more efficient algorithm. As illustrated in Figure 2, AlphaZero allows game agents to self-play deterministically against only the latest policy, a method known as Myopic BR. This approach encourages agents to learn myopically to minimize their immediate strategy's maximum loss. However, this method may not be effective in all games. For example, in a game like Rock, Paper, Scissors, self-play may cycle myopically and not ensure a NE strategy interaction.

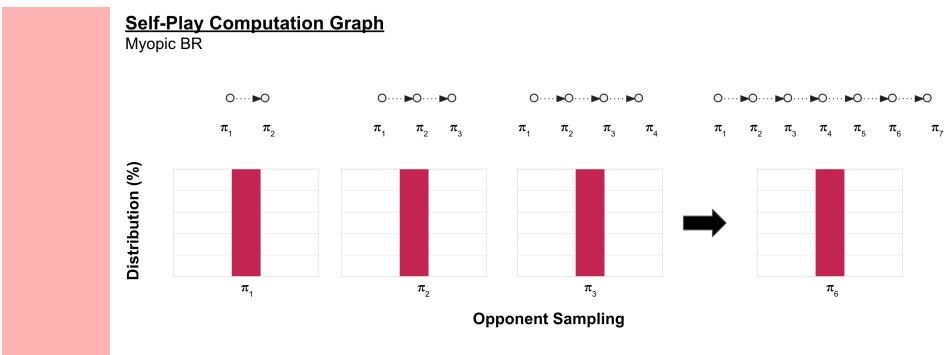

Figure 2: An illustration of Myopic BR - AlphaZero allows game agents to self-play deterministically against only the latest policy, a method known as Myopic Best Response (Myopic BR).

This dichotomy between theoretical guarantee of convergence and efficiency of training presents a gap in the current understanding of self-play algorithms. Our study aims to bridge this gap by understanding how cycles of strategy interactions relate to a NE, and whether we can find a Nash equilibrium more efficiently, regardless of the presence or absence of cycles. We establish the theoretical equivalency between a complete set of cyclical best response strategies and the support set of a MSNE. Our proof shows that the cyclical best response strategies must form a circular counter, implying that a complete set is necessary to support an MSNE due to the intrinsic counterbalancing dynamic. This means that AlphaZero's efficient Myopic BR self-play will either transitively converge to a PSNE or find a sequence of cyclical policies that has an intrinsically counterbalance as the support set of an MSNE. This allows us to represent the learning representation of an equilibrium point in noncooperative games as a directed graph search.

We introduce Graph-based Nash Equilibrium Search (GraphNES), a novel self-play algorithm that enhances Myopic BR by identifying if and when the self-play sequence has developed a cycle that repeats over prior strategy interactions. GraphNES allows agents to calculate BR over fewer opponent strategies, thereby more efficiently identifying a PSNE or the supports of a MSNE in noncooperative games. We empirically evaluate our approach on various noncooperative games, including Connect4 and Naruto Mobile. Our results show that our graph-based learning representation significantly improves the efficiency of finding an equilibrium point in self-play compared to previous methods.

## 2 BACKGROUND

In this section, we review the previous works on finding NE in noncooperative games and their underlying assumptions and methods.

**Game Theory Basics**: A noncooperative game can be represented as a normal-form matrix game of tuple $(n, M, U)$, where n is the number of players, $ps^{id}$ is the set of pure strategies for player $id \in n$, $M = (ps^1 \times, \ldots, ps^n)$ is the matrix of all possible strategy combinations, and $U := M \to R^n$ is the payoff function that maps each combination to a vector of payoffs for each player. The players aim to maximize their expected utility by choosing either a pure strategy or a mixed strategy, which is a probability distribution over $ps^{id}, \sigma^{id} = ((p_1 \times ps_1^{id}), \ldots, (p_N \times ps_N^{id}))$, where N is the size of the strategy set and $p_c$ is the probability of each strategy with $\sum_{c=1}^N p_c = 1$. The support set of a MSNE is the set of pure strategies that are played by the players with positive probability in the equilibrium. It ensures that each player's mixed strategy is optimal and cannot be improved upon by playing a different pure strategy. For games with known payoff values and small matrix size, linear programming (LP) can be used to solve for support set of NE. However, LP is not applicable for games with unknown or large payoff matrices, such as Go (Silver et al., 2016).

**Deep Reinforcement Learning (DRL)**: To deal with the unknown payoff values, DRL (Sutton & Barto, 2018) is used to learn and estimate the expected return of each agent's policy. The agent updates its policy based on the experience tuple $(O^{id}, A^{id}, R^{id})$, where $O^{id}$, $A^{id}$ and $R^{id}$ are the agent's observation space, action space and accumulated return respectively. The discounted return at time step $t$ is defined as $R_t^{id} = \sum_{\tau=t}^{\infty} \gamma^{\tau} r_{\tau}$, where $\gamma$ is the discount factor for all agents in the range $[0, 1)$.

**Fictitious Self-Play (FSP)**: FSP (Heinrich et al., 2015) extends the return estimation to the opponent's action sequence in alternating turns of self-play. DRL and FSP together enable the agents to learn from unknown payoff values in noncooperative games.

**Iterated Best Response (IBR)**: To handle the large matrix size, IBR divides the matrix game into subgames $m_k \in M$ for $k \in K$ self-play rounds. The agents start with a subset of policies and iteratively add new best response (BR) policies $\{\Pi_k^{id}\}_{id=1}^n = (\pi_k^1, \ldots, \pi_k^n)$ against their previous policies in each subgame. This makes the NE computation tractable over a large matrix game.

**Best Response Dynamics**: IBR can be learned with respect to all previous opponent policies, which is called best response dynamics, or with respect to only the latest opponent policy, which is called Myopic BR. Myopic BR is more efficient and effective in finding improved counter policies, as it only considers the most updated opponent policy. However, Myopic BR may result in a cycle of policies that keep switching without reaching a NE. This is called a cyclical strategy.

**Cyclical Strategy**: In simultaneous move or imperfect information games, the learning process under Myopic BR may cycle back to a previous policy in the self-play sequence, resulting in repeated self-play and potentially failing to converge to a NE. This phenomenon, known as cyclical strategy, is discussed in (Balduzzi et al., 2019). For example, in the game of Rock, Paper, Scissors, the strategies may cycle as Rock $\Rightarrow$ Paper $\Rightarrow$ Scissors $\Rightarrow$ Rock, and so on. To prevent this, existing approaches encourage players to learn the best response against all previous sequences of policies $(\Pi_{[0:1]}, \Pi_{[0:2]}, \ldots, \Pi_{[0:k-1]})$ for $k \in K$ self-play rounds.

**Current Research Methods**: Current research methods, such as PSRO, OpenAIFive, and Simplex-NeuPL (Liu et al., 2022), prioritize the sample distribution of the opponent population (e.g., softmax, Dirichlet sampling distribution on opponents' win rates). This allows an agent to learn against the optimal probabilistic play of the opponent mixed strategy $(\sigma_{[0:1]}, \sigma_{[0:2]}, \ldots, \sigma_{[0:k-1]})$. The effec-

tiveness of these methods is guaranteed by the Double Oracle method and (Roughgarden, 2010), which state that the best response dynamics of self-play sequence must converge to a NE in any finite noncooperative game. These advancements have significantly improved our ability to find NE in complex games, but at the cost of intensive compute.

# 3 LITERATURE REVIEW

The study of noncooperative games, particularly their representation as graphs and the identification of Nash equilibria on such graphs, has been a significant area of study in academic literature. However, the existing body of work presents a dichotomy.

We begin with the foundational work of Nash (1951), which asserts that every finite game has at least one Nash equilibrium, potentially in mixed strategies, using Kakutani's fixed point theorem (Nash Jr, 1950). This theorem, based on set-valued functions where one input can map to a set of outputs, ensures the existence of a fixed point. This suggests that even if one strategy has a set of best responses, a Nash equilibrium can still exist.

Building on this, Vlatakis-Gkaragkounis et al. (2020) examined the relationship between no-regret learning and MSNE (Vlatakis-Gkaragkounis et al., 2020). They established that any Nash equilibrium that is not strict (where every player has a unique best response) cannot be stable under no-regret learning. This finding contradicts Nash's classical result.

In response to this contradiction, Milionis et al. (2022) demonstrated that certain games exist where any continuous or discrete time dynamics fail to converge to a Nash equilibrium (Milionis et al., 2022). They argued that the Nash equilibrium concept is affected by a form of incompleteness that allows cycling, and suggested that the problem is intractable, advocating for the exploration of alternative equilibrium concepts.

Further exploring the concept of cycles, Mertikopoulos et al. (2018) scrutinized the cycling behavior of adversarial regularized learning (Mertikopoulos et al., 2018). They revealed that rapid regret minimization does not necessarily lead to swift equilibration, and that cycles can emerge even in straightforward two-player games. However, they did not propose a method to identify a Nash equilibrium when a cycle is encountered.

Contrary to these views, there are works that argue that cycles can, in fact, aid in finding a Nash equilibrium. One of the earliest works that discussed the role of cycles in finding a Nash equilibrium is by E. Akin, who showed that the set of strongly connected components of the best response graph contains the support of a mixed Nash equilibrium (Akin, 1980).

Another work that explored the connection between cycles and Nash equilibria is by Biggar and Shames, who introduced the concept of chain components and the response graph (Biggar & Shames, 2023). They proved that every chain component is a strongly connected component of the best response graph, and that every Nash equilibrium corresponds to a fixed point of the response graph.

Finally, we discuss Daskalakis et al.'s End of the Line argument, which highlights the PPAD hardness (Daskalakis et al., 2009). The primary argument of End of the Line is to represent a noncooperative game as a directed graph, thereby equating the discovery of a Nash Equilibrium to finding a fixed point of a graph's leaf or a cycle.

However, these works have yet to show how many set of strongly connected components are needed to support a MSNE, or if any MSNE of any finite game as indicated by Nash's original proof in 1951 are strongly connected components on a graph. To make such a strong claim, we must prove an 'equivalence' relation between the support set of a mixed strategy Nash equilibrium and the complete set of cyclical strategy. This is the motivation and contribution of our paper. We aim to bridge this gap in the literature by providing a robust method for managing cycles and identifying Nash equilibria in such scenarios.

## 4 PROOF OF EQUIVALENCY

To address these challenges, we need to understand the relationship between cyclical strategies and NE. In the following, we show that a complete set of cyclical BR strategies is equivalent to the support set of a MSNE.

**Theorem 4.1.** *A complete set of cyclical BR strategies is equivalent to the support set of a MSNE.*

Before we delve into the proof, let's clarify some definitions and notations:

- A MSNE is a state of strategic interaction where no player can unilaterally deviate from their probabilistic play of their support strategies to increase their expected payoff. This is in contrast to a PSNE, where players choose their strategies deterministically such that the strategy interactions form a NE.

- The expected payoff of player $id$ using pure strategy $ps_1^{id}$ against pure strategy $ps_2^{-id}$ of the opponent is denoted as $U(ps_1^{id}, ps_2^{-id})$.

- A pure strategy $ps_i$ is a BR to another pure strategy $ps_j$ if it maximizes the payoff among all possible pure strategies, i.e., $U(ps_i, ps_j) \geq U(ps_{\hat{i}}, ps_j)$ for all $\hat{i}$.

- A cyclical set is a collection of strategies where each strategy is a BR to the previous strategy in the set.

- A cyclical set $C$ is complete if adding any other pure strategy to it does not improve the payoff of any mixed strategy with support in $C$. Formally, if $\sigma_*^{id}$ is the optimal mixed strategy for player $id$ with support in $C$, and $\sigma_{\hat{*}}^{id}$ is the optimal mixed strategy that includes the pure strategy of $ps_c^{id}$ with support in $C' = C \cup ps_c^{id}$, then $U(\sigma_*^{id}, \sigma_*^{-id}) \geq U(\sigma_{\hat{*}}^{id}, \sigma_*^{-id})$, where $\sigma_*^{-id}$ is the optimal mixed strategy of the opponent.

With these definitions and notations in mind, we can now proceed to the proof of the theorem.

*Proof.* Individual player's support strategies of a MSNE together as an union set is a complete set of cyclical BR strategies.

We first show players' probabilistic play of their individual support strategies must union together that forms a complete set of cyclical BR strategies. The union of the support strategies helps to illustrate why players must play their support strategies probabilistically rather than deterministically.

1. **Definitions**: Let $\sigma_*^{id} = (p_1 \times ps_1^{id}, \ldots, p_s \times ps_s^{id})$ denote the MSNE mixed strategy for player $id$, where $\{ps_c^{id}\}_{c=1}^s$ is the support set of pure strategies with $\sum_{c=1}^s p_c = 1$. Let $SS$ represent the union support set for a MSNE for all players.

2. **Assumption**: Suppose that there exists a modification of $\sigma_*^{id}$ of player $id$'s mixed strategy such that a probability of a support strategy $p_j$ for some $j \in s$ is decreased to deviate from a MSNE, and that no opponent $-id$ may play more of their pure strategy in $SS$ to increase their payoff. That is, let $\sigma^{id}$ be the modified mixed strategy that is not a MSNE such that $U(\sigma^{id}, \sigma_*^{-id}) \text{ ¡ } U(\sigma_*^{id}, \sigma_*^{-id})$ for player $id$, and $U(\sigma_{\hat{*}}^{-id}, \sigma^{id}) \leq U(\sigma_*^{-id}, \sigma^{id})$ for any $\sigma_{\hat{*}}^{-id}$ with the support set in $SS$.

3. **Contradiction**: If the decrease of $p_j$ for some $j \in s$ does not lead to an increase of an opponent's BR strategy in $SS$, then player $id$ must be able to increase $p_j$ to increase their payoff. However, this contradict that $\sigma_*^{id}$ is a MSNE for player $id$. Moreover, an opponent $-id$ must be able to increase the probability of one of its support strategies in $SS$ to exploit the decreased probability of $p_j$ for the support strategy $ps_j^{id}$. This is because the opponent's mixed strategy is also optimal and has a BR to every strategy in $SS$. Otherwise, the strategy $ps_j^{id}$ is not a BR to any opponent's strategy that is currently in play. This contradicts our assumption that $U(\sigma_*^{-id}, \sigma^{id}) \leq U(\sigma_*^{-id}, \sigma^{id})$ for any $\sigma_{\hat{*}}^{-id}$ with the support set in $SS$. Therefore, every player's support strategy must have a BR counter by an opponent that is in the set $SS$.

4. **Assumption**: Next, suppose that it is not possible to order $SS$ into a sequence of cyclical BR.

5. **Contradiction**: However, this contradicts what we have proved that every player's support strategy must have a BR counter that is in the set $SS$.

6. **Conclusion**: Since the contradiction apply to all players' mixed strategy of a MSNE, and by the definition of MSNE $SS$ must be a complete set. Hence, we conclude that if $SS$ is the union of support strategies of a MSNE then it must be cyclical and complete.

*Proof.* A complete set of cyclical BR strategies is also players' union support strategies of a MSNE.

1. **Definitions**: Let $CS = (ps_1^{id}, ps_2^{-id}, \ldots, ps_k^{-id})$ denote a sequence of Cyclical BR Strategies of the participating players, and let $CS$ be a complete set based on the defined definition. Players may use their pure strategies in $CS$ to compose a mixed strategy.

2. **Assumption**: Suppose that it is not possible for player $id$ to play a mixed strategy $\sigma_*^{id}$ with with its pure strategies from $CS$ to support a MSNE. This means that there exists a mixed strategy $\sigma_{\hat{*}}^{id}$ with a different union support set of $CS'$ than $\sigma_*^{id}$'s union support set of $CS$ such that the expected payoff of $\sigma_{\hat{*}}^{id}$ is strictly higher than $\sigma_*^{id}$ when played against the opponents' optimal mixed strategy $\sigma_*^{-id}$. i.e., $U(\sigma_{\hat{*}}^{id}, \sigma_*^{-id}) > U(\sigma_*^{id}, \sigma_*^{-id})$.

3. **Contradiction**: Given that $CS$ is a complete set, a higher payoff of $\sigma_{\hat{*}}^{id}$ must have an equal or smaller support set size than $\sigma_*^{id}$. This means $\sigma_{\hat{*}}^{id}$'s corresponding union support set of $CS'$ must has a equal or smaller union support set than $CS$. However, since $CS$ is cyclical, each strategy in $CS$ is BR to the preceding strategy in the set. Removing any support strategy $ps_i^{id}$ from $CS$ would result in player of $id$ of mixed strategy $\sigma_*^{id}$ in an equal or lower payoff. This means $CS'$ must have equal size of union support set to $CS$.

Suppose $CS'$ has the same size of union support set to $CS$, changing any player's mixed strategy support of a pure strategy from $ps_i^{id}$ to $ps_j^{id}$ would not increase player $id$'s payoff since strategy $ps_i^{id}$ is already a BR payoff of $ps_{i-1}^{-id}$. This contradicts our assumption that $\sigma_{\hat{*}}^{id}$ has a strictly higher payoff than $\sigma_*^{id}$.

4. **Conclusion**: Since the contradiction of assumption may be applied to every player and their support strategies, our assumption that $CS$ is not a union support set of a MSNE must be false and we conclude that a complete set of cyclical strategies must support a MSNE.

Thus, we have shown that if $SS$ supports a MSNE then it must be equivalent to $CS$, and vice versa.

This completes our proof. We show an example of (Rock,Paper,Scissors,Lizard,Spock) to illustrate why complete set is a necessary condition in Appendix [**A.1**]

## 5 METHODOLOGY

In this section, we introduce Graph based Nash Equilibrium Search (GraphNES), a novel self-play algorithm that uses a directed graph representation to find a PSNE or a MSNE in noncooperative games. Based on Theorem 4.1, the Myopic BR self-play will either converge to a PSNE, or identify a cycle of BR policies that form the SS of a MSNE. In both cases, finding a PSNE or the SS of MSNE can be reduced to a graph cycle detection problem. We first explain the graph representation in Figure 3.

**Myopic BR Self-play Sequence:** We explore $G(V, E)$ by starting with an initial vertex $v_0$ representing an agent's initial policy, and perform Myopic BR self-play as one step of depth first search to expand the latest vertex $v_0^{-id}$ to $v_0^{-id} \Rightarrow v_1^{id}$. Similarly, the iterative Myopic BR self-play sequence continues to expand the latest vertex $v_{k-1}^{-id}$ to find a neighboring vertex $v_k^{id}$ with a BR payoff gain for the next agent of $id$.

Myopic BR's self-play sequence either converges with $v_k^{id}$ that cannot expand further as a PSNE, or an ongoing sequence of BR policies $[v_0^{-id} \Rightarrow \ldots \Rightarrow v_k^{id}]_{id=1}^n$ with potential of cycle.

**Identify Cycle and Support Set:** To identify if the current vertex $v_k^{id}$ has formed a cycle with a prior vertex in $(v_0^{id}, \ldots, v_{k-1}^{id})$, we compare their strategy similarities. We record agents' policies $\pi_{\theta_{[0:k]}}(\cdot|o_t, id)_{id=1}^n$ and their distinct Myopic BR action sequence of $[A_0, \ldots, A_k]$ during self-play. We let $[A_0, \ldots, A_k]$ denote the 'colors' of each vertex. We query $A_k^{id}$ to search for similar color characteristic in the set of $(A_0^{id}, \ldots, A_{k-1}^{id})$ with a vector database search.

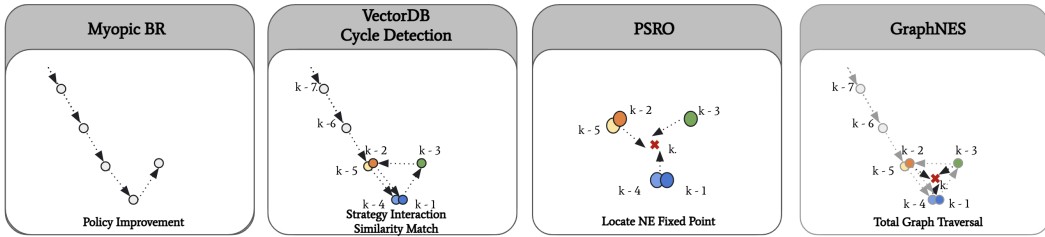

Figure 3: **Graph Representation:** We construct a directed graph $G(V, E)$ for a noncooperative game. Each vertex $v \in V$ represents a unique pure strategy of a player. Each edge $e \in E$ represents a BR policy of another player with a positive payoff gain. Formally, $v_{k-1} \Rightarrow v_k$ means $v_k$ is a BR policy of $v_{k-1}$ with a positive payoff gain. Since $E$ are defined by BR policies with positive payoff gain, $G(V, E)$ has no self-loop. Finding a NE is equivalent to finding either a vertex with no outgoing edge PSNE, or a set of vertices that form a complete cycle (support set, SS, of a MSNE).

---

**Def identifyCycle**(memoryInteract):
1: DB.build(memoryInteract[0:-2]) *Build a vector database
2: $(x, \text{dist}) \leftarrow$ DB.query(memoryInteract[-1])
3: **if** (dist ¡ distLow) Then **then**
4:    distLow = Dist
5:    DC = $EVAL(v_{x+1}^{-id}, v_k^{id})$ *Double Check
6:    **if** (DC) Then **then**
7:       cycleDetect $\leftarrow$ True
8:       supportPolicy $\leftarrow v_{((x:k])}$
9:    **end if**
10: **end if**
11: Output cycleDetect, supportPolicy

---

For $A_x^{id} \in A_{[0:k-1]}^{id}$ such that $A_x^{id}$ is the most similar match of $A_k^{id}$, we mark the potential $v_k^{id} = v_x^{id}$ as a repeat of policy due to cycle. Observe if $v_k^{id} = v_x^{id}$, then $v_{x+1}^{-id}$ must be a BR to $v_k^{id}$ such that $v_k^{id} \Rightarrow v_{x+1}^{-id}$. Thus, we may verify the color match by evaluating $v_{x+1}^{-id}$ against the current policy $v_k^{id}$. If $v_{x+1}^{-id}$ is a BR against $v_k^{id}$, $\pi_{\theta_{(x:k]}}(\cdot|o_t, id)_{id=1}^n$ is a sequence of cyclical BR policies $CS_0$. Otherwise, continue Myopic BR to expand the self-play sequence.

---

**Def GraphNES**($\pi_\theta$, cycleDetect, supportPolicy, $v_{k-1}^{-id}$, k):
1: **if** (not cycleDetect) **then**
2:    $A_k$ = MyopicBR($\pi_\theta$, $v_{k-1}^{-id}$)
3:    memoryInteract = memoryInteract $U$ $A_k$
4:    cycleDetect, supportPolicy $\leftarrow$ identifyCycle(memoryInteract)
5:    k = k + 1
6:    $v_{k-1}^{-id} = \pi_\theta$
7: **else**
8:    PSRO($\pi_\theta$, {supportPolicy}$_{id=1}^n$)
9:    supportPolicy = supportPolicy $U$ $\pi_\theta$
10: **end if**
11: Output ($\pi_\theta$, cycleDetect, supportPolicy, $v_{k-1}^{-id}$, k)

---

**Completing the Set:** To evaluate and complete the set $CS_0$, we utilize PSRO to find higher payoff BR policy outside of $CS_0$. If $CS_0$ is already a complete set, PSRO immediately find the probabilistic mixtures and converge to the MSNE policies for each agent. Else, PSRO iteratively adds new BR policies to $CS_i$ to complete the set. Let $CS_i = CS_{i-1}$ $U$ $\pi_{\theta_{[k+1]}}(\cdot|o_t, id)_{id=1}^n$. Note that due to neural net policies are approximate functions, the support found may be approximation. We denote

the result of self-play convergence as $\epsilon-$ NE, where $\epsilon$ is the difference of payoff from the found players' strategies to the actual NE of the game.

# 6 EXPERIMENT

In this section, we conduct empirical studies on two types of noncooperative games to evaluate our method. We use Connect4 as an example of a two-player, turn-based, perfect-information game that illustrates a purely transitive game. We use Naruto Mobile as an example of a simultaneous move game, where finding a NE involves GraphNES mapping the self-play sequence to both transitive and cyclical policies. We compare our method with PSRO and Simplex-NeuPL. We use identical settings, such as hyperparameters, model architecture, and so on. The only difference between the methods is the self-play algorithm employed.

## 6.1 CONNECT4

Connect4 is a game played on a 6x7 grid where two players compete to be the first to align four of their colored pieces in a straight line, either vertically, horizontally, or diagonally. Players take turns placing their pieces until the game ends. Connect4 has approximately 4.5 trillion possible move positions, which is less complex than Go but still challenging.

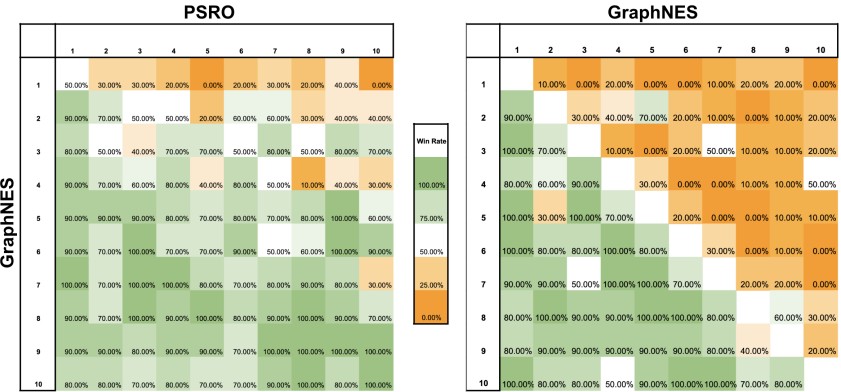

Figure 4: We trained two agents using GraphNES and PSRO to learn Connect4, as shown in the **left** figure. We terminated each training round once an agent found a BR policy against its self-play opponents. This design measures the efficiency of each method in finding a BR with the same computation time per round. We then evaluated 20 policy models as agents' strategies in a competition. We also studied the convergence of GraphNES's self-play sequence, as shown in the **right** figure. All win rates displayed are for the row players.

Figure 4 shows that both methods performed similarly in the first three rounds of policy optimization. However, GraphNES improved its policy performance faster than PSRO as the self-play sequence continued. In another experiment, GraphNES showed a transitive policy improvement with no BR cycle, as shown in the right figure.

These experiments suggest that an agent can learn better by using Myopic BR to improve its latest policy, allowing it to find and fix errors until reaching a PSNE. This is consistent with Zermelo's Theorem (Zermelo, 1913), that Connect4 shares four common properties with Go, Chess, and Shogi: they are two-player, turn-based, perfect-information games with a finite number of moves. According to Zermelo's Theorem, such deterministic games have deterministic equilibrium strategy interactions, i.e., PSNE. This further supports the effectiveness of the Myopic BR approach in these deterministic games.

## 6.2 NARUTO MOBILE

Naruto Mobile is a popular Fighting Game that features simultaneous-turn gameplay, similar to Street Fighter. In this game, players compete to maintain higher health points (HP) than their op-

ponents within a limited time. The game offers a variety of characters, each with unique skills and attributes such as movement speed and body size. Players aim to maximize their win rate by strategically using these skills, which can be used for attack, evasion, area of effect, or status change. Since players act at the same time, this game poses a challenge for finding cyclical strategies and MSNE.

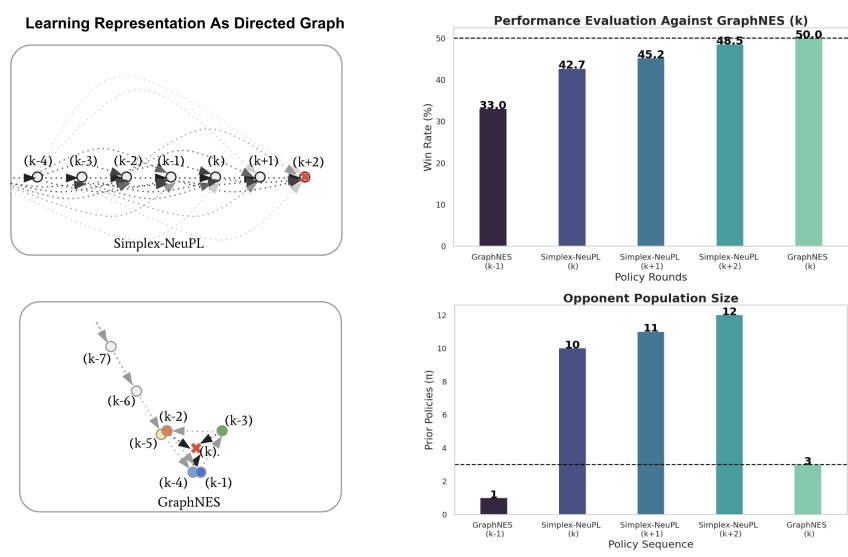

Figure 5: Simplex-NeuPL follows a best response dynamic training, where agents' self-play sequence learns to develop new policies to best respond against a symmetric Dirichlet distribution of previous policies. The learning converges approximately at round 13. In contrast, GraphNES's Myopic BR does not encounter a cycle of BR until the 9th round, and detects the cycle on the 10th.

In Figure 5, Simplex-NeuPL performed a total count of 1 + 2 + ... + 12 = 78 best responses against GraphNES's 9 Myopic BR, and a PSRO of the three colored trio of strategies of 3 on the 11th round. The total self-play opponent count of GraphNES is only 9 + 3 = 12 over the self-play sequence. The self-play simulation efficiency demonstrates a gain from 78 to 12 with a 6.5x improvement. The more efficient self-play also translate to a more optimal approximation of MSNE with GraphNES outperforms Simplex-NeuPL's round k by 7.3 %, round k+1 by 4.8 % and round k + 2 by 1.5 % of win rates. The experiment suggests that self-play against more diverse but weak opponents is less efficient than self-play against the support set of an approximate MSNE.

## 7 CONCLUSION

In this study, we investigated the relationship between probabilistic behaviors, cycles, and equilibrium in agents. We found that players' strategy interactions in any finite noncooperative game either converge to a deterministic PSNE through transitive improvement, or form a cycle(s) of an MSNE that requires probabilistic play.

Our findings help improve the computational efficiency of self-play algorithms and connect the noncooperative game learning representation to a directed graph search. This representation helps identify cycles in a potentially large set of strategy interactions and explains why agents benefit from having memory that can recognize recurring patterns of interactions.

**Limitation**: One limitation of our approach that prevents scaling to more complex environments such as StarCraft or the real world dynamics is the vector database storage and query. When a single strategy is very long, the vector that represents it can become too sparse to be retrieved accurately. In future research, it may be possible to use algorithms such as Dynamic Time Warping to handle long-term temporal strategy interactions. We discuss this challenge in Appendix [**A.3**]

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
