## A    APPENDIX

### A.1    ILLUSTRATION OF WHY 'COMPLETE' SET OF CYCLICAL BR IS NECESSARY

Here we dive in an example of the game (Rock, Paper, Scissors, Lizard, Spock) to demonstrate the reason why finding a single set of cyclical BR may not be sufficient to ensure a MSNE of counterbalance. Invented by Sam Kass and Karen Beryl (Kass, 1995), and popularized by the TV show The Big Bang Theory, the noncooperative game of (Rock, Paper, Scissors, Lizard, Spock) is a simultaneous-turn game that expands the original triangular cycle of (Rock, Paper, Scissors) to a 5-cell simplex. The game has the following rules:

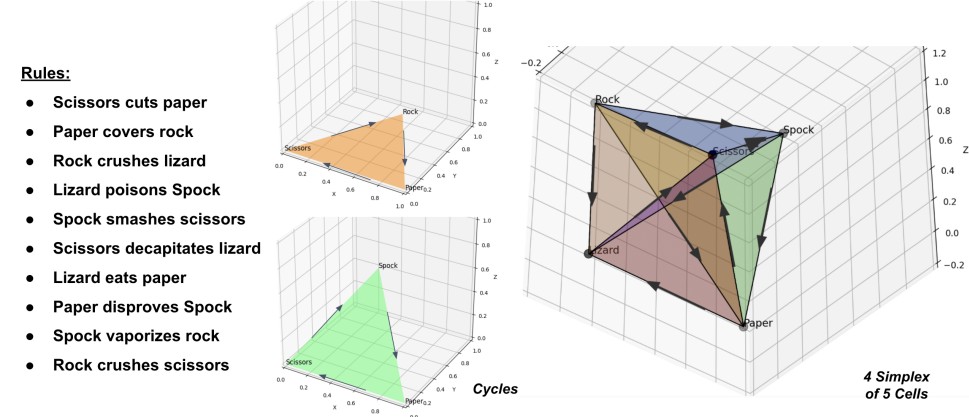

Figure 4: The game we're discussing can be visualized as a graph of a 4 Simplex with 5 Cells, where each cell represents a pure strategy: (Rock, Paper, Scissors, Lizard, and Spock). The game is characterized by multiple cyclical BR sequences, such as (Rock⇒ Paper⇒ Scissors⇒ Rock. . . ), (Paper⇒ Scissors⇒ Spock⇒ Paper. . . ). Each sequence forms a counterbalance among the pure strategies.

From Figure 4 let's take a closer look at one of these cycles: (Rock⇒Paper⇒Scissors⇒Rock. . . ). If the players are only allowed to play these three strategies, an equilibrium would be achieved with a uniform distribution of the three strategies. However, if a player can also play Spock in a new game of (Rock,Paper,Scissors,Spock), the cyclical counterbalance shifts. Since Spock is the BR against both Rock and Scissors but is beaten by Paper, the new cycle becomes (Scissors⇒Rock⇒Spock⇒Paper⇒Scissors...). The MSNE remains cyclical and counterbalanced, but the probability weighting of these strategies changes.

If we expand the total strategies to include all five options (Rock, Paper, Scissors, Lizard, Spock), the MSNE support set becomes a complete cyclical BR sequence of (Rock⇒ Paper⇒ Scissors⇒ Spock⇒ Lizard⇒ Rock. . . ). Each strategy in this set is counterbalanced by another strategy. This results in an MSNE for the game of $(\frac{1}{5}Rock, \frac{1}{5}Paper, \frac{1}{5}Scissors, \frac{1}{5}Lizard, \frac{1}{5}Spock)$ for all players. This demonstrates how a complete set of cyclical strategies continues to be part of the support set of the equilibrium strategy interactions of the game.

### A.1.1    NONPLANAR GRAPH VERSUS n-DIMENSIONAL GRAPH - TOPOLOGY OBJECTS

In this section, we explore the distinction between Nonplanar Graphs and n-Dimensional Graphs (Topology Objects), using the game Rock-Paper-Scissors-Lizard-Spock as a case study.

The left figure on Figure 5 shows the online representations depict of the game as a Pentatope of $K_5$, known in Graph Theory as a nonplanar graph. Nonplanar graphs are characterized by *edge intersections* beyond their endpoints. However, we propose that the game's graph actually exists in a higher dimension - as a 4-Simplex.

The reason behind this assertion lies in the interpretation of nonplanar graphs like $K_5$. Such graphs indicate conflicts in vector directions, represented by edge intersections. This implies a disagree-

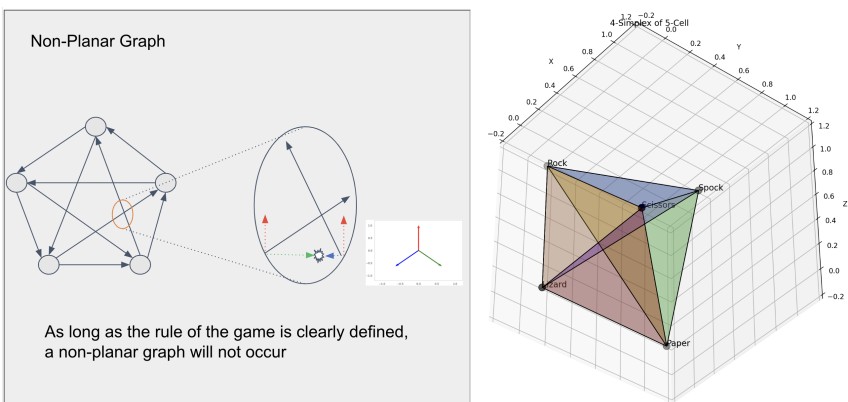

Figure 5: The game is commonly represented by a graph where each vertex, indicating a pure strategy, has two incoming and two outgoing edges. These edges point to the BR of that strategy. This 2D representation simplifies the understanding of the game mechanics. However, it's crucial to note that the well-defined rules of the game do not allow for intersecting edges in its graph. Instead, the game's graph can be more accurately expressed as a 4-Simplex of Graph Topology Object.

ment between players on the outcome of two strategies' interaction - each player believes they've won. However, in Rock-Paper-Scissors-Lizard-Spock, the rules clearly define the winner, thereby eliminating any possibility of such conflicts or edge intersections.

To summarize, non-planar graphs typically occur in games with ill-defined rules leading to edge intersections at points other than their endpoints. On the other hand, games with well-defined rules result in either planar graphs or n-dimensional graph topology objects. A well-defined set of rules ensures that all players agree on game outcomes - whether it's a win, loss or tie."

### A.1.2 GRAPH LEARNING REPRESENTATION

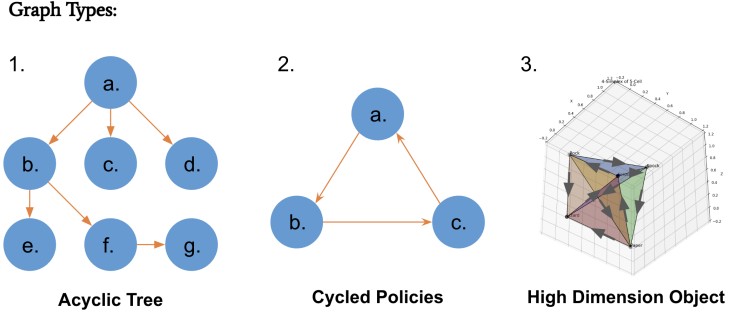

Figure 6: We can categorize the graph search for any noncooperative game of clearly defined rules into three cases. First, we consider the case where a game has a deterministic PSNE. The directed graph search can be represented as an acyclic path. Second and third, we consider the case where a game has a nondeterministic MSNE, and the graph representation can be either exist in a 2D plane or n-dimensional.

In Figure 6 we may visualize the three types of graph learning representation for any noncooperative game of clearly defined rules. Observe that the three cases represent a typical increase of simplex dimensions. In the case of PSNE, players only play a single pure strategy. This may be represented as a single point of a simplex. In the case of a single cycle, this would be a simplex of 1 face. We may further extend to n-dimensions demonstrated in (Rock,Paper,Scissors,Lizard,Spock), where a game may have multiple cycles and therefore multiple graph faces (4-Simplex 5 Cells).

These faces forms the surface of a topology object. Hence the research in graph and topology are often jointly studied as a single research discipline.

### A.1.3 How Fixed Points (NE) Exist in A Graph Topology Representation

We may look at the following three fixed point theorems (Brouwer, 1912),(Daskalakis et al., 2009) and (Schauder, 1930) to understand why Nash based his proof of equilibrium point in noncooperative games on Fixed Point Theorem.

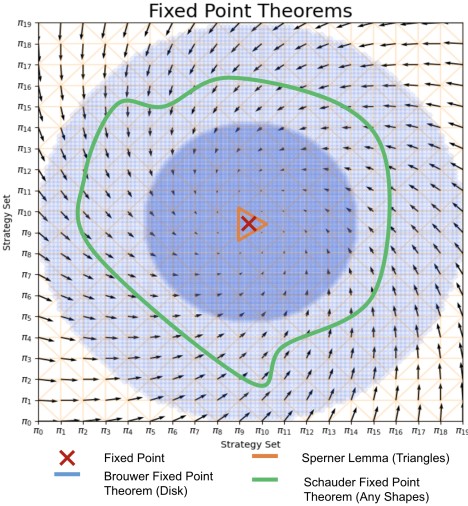

Figure 7: The figure shows a 2D plot of different Fixed Point Theorems. The existence of a fixed point, or a Nash equilibrium (NE) in Game Theory, was first proven using the Brouwer Fixed Point Theorem for disk and ball objects, where a continuous mapping of a compact region of light blue to itself, blue, is guaranteed to exist at least one fixed point. Later shown by Daskalakis et al., by triangulating the whole graph Sperner's Lemma also guarantees a NE fixed point within triangle and simplex objects of complete cycle of change of edge directions. For the general case of Banach space of infinite dimensions, Schauder Fixed Point Theorem (Schauder, 1930) proves the existence of a fixed point in any dimension of graph convex hull.

Looking at the three Fixed Point Theorems in Figure 7, observe that a fixed point may fall within a complete cycle of disk, triangle or any shape of graph cycle. Intuitively this is due to a graph cycle having a continuous change of edge directions that cycle back to itself. This means that within any graph cycle, there exists at least one point that has change of directions sum to zero. Fixed Point Theorems also suggest fixed points may exist in n-dimension topology objects such as ball, simplex.

## A.2 HYPERPARAMETERS AND HARDWARE USED

### A.2.1 CONNECT4

Hyperparameters

- MCTS SIMS = 15
- EPSILON = 0.2
- Reward discount factor: 0.995
- BATCH SIZE = 128
- LR = 0.1
- MOMENTUM = 0.9
- EVAL GAMES = 20

Hardware Used

- CPUS: 1
- GPUS: 1

### A.2.2 NARUTO MOBILE

Hyperparameters

- PPO: 0.1
- n-step: 100 frames
- Reward discount factor: 0.995
- LR: 1e-4
- BATCH SIZE: 300
- MOMENTUM = 0.9
- EVAL GAMES = 300 - 500

Hardware Used

- CPUS: 5,000
- GPUS: 10

## A.3 VECTORDB

- Index Type = HNSW
- Batch Size Until DB Rebuild = 1

In Naruto Mobile where each agent control a set of characters to learn the equilibrium point of the character strategy interactions, we use the following metrics to record and identify cycle:

- forcingMove (Characters' actions that initiating an engage)
- counterMove (Characters' actions that counteracting an opponent's forcingMove)
- hitRate (An attack that result in damaging the opponent)
- evadeRate (1 - hitRate)
- itemUse

We measure the expected metric values of the controlled character set to use them to represent the strategies (action-sequence) of agent's policies. These metrics are recorded as memoryInteract and stored into a vector database for query and retrieval. Query and retrieval of high similarity of policies helps to identify if and when the current policy may have cycled and learned a repeat of a prior policy. Identify these similar policies is analogous to identify the *same colored vertices* in a typical Depth First Search Algorithm of identifying a cycle on a directed graph.

For other games that has a relative short action sequences, such as Connect4, a policy's output of action sequence may be used as the vectors to record and store. For longer action-sequence strategies such as StarCraft, the long vector may result in a sparse retrieval. A Sparse retrieval means there are too many actions in a single strategy, and record and query of these vectors would lead to a poor retrieval result since every pair of vector distance would be very large. To the best of our knowledge, this was where the problem of Curse of Dimensionality was originally coined where the long vector dimension hinders the accuracy of the retrieval result.

To tackle this problem, we believe the approach of Dynamic Time Warping would be necessary as we mentioned in the Conclusion (limitation) of our study. This would breakdown the long sequence of vectors into smaller dimension(s) and match them sequentially using dynamic programming.

## A.4 MISCELLANEOUS

### A.4.1 SOFTWARE AND LICENSING

The algorithm are implemented via Tensorflow (Abadi et al., 2015), IMPALA (Espeholt et al., 2018), and Horovod(Sergeev & Balso, 2018).
These softwares are all licensed under Apache License 2.0.