# OpenReview forum: "The Cyclical Chaos And Its Equilibrium"
_ICLR.cc/2024/Conference — Submitted to ICLR 2024_

### Official Review · Reviewer_Sw3W · 2023-10-24

**Soundness:** 3 good
**Presentation:** 3 good
**Contribution:** 2 fair
**Rating:** 3
**Confidence:** 3

**Summary:**

The paper considers iterative methods to determine Nash equilibria in finite, non-cooperative games. To avoid cycles of best responses, e.g., as is the case with myopic best response in Rock-Paper-Scissors, current state-of-the-art iterative methods like PSRO calculate best responses against all previous policies. This is effective in avoiding cycles, leading ultimately to learning a Nash equilibrium, however, it has a considerable (and often forbidding) increase in computational time by continuously increasing the opponent's self-play population of policies.

With the aim to improve the self-play algorithm complexity, the paper proposes the idea that when such iterative algorithms enter a cycle, then this cycle must be the support of a mixed Nash equilibrium. It provides a rigorous proof that a complete cyclical set is necessary and sufficient to form the support of a Nash equilibrium. This implies that myopic algorithms, like AlphaZero, can in fact learn the support of a Nash equilibrium when they enter a cycle (or of course, learn a pure Nash equilibrium). Based on this intuition, the paper proposes a self-play algorithm, called GraphNES, that exploits the graph-search dynamics of the above formulation to either stop at a pure Nash equilibrium or to identify that it has entered a cycle. Empirical evaluations on Connect4 and Naruto Mobile suggest improved performance over baselines.

**Strengths:**

- The paper has a solid motivation: finding iterative self-play algorithms with low sample complexity is an interesting open problem.
- The paper contains a rigorous result (Theorem 3.1) and interesting experiments in two games, Connect4 and Naruto Mobile.
- Some parts of the paper are clearly written, e.g., the introduction, and allow the reader to understand the context and claimed contributions of the paper.

**Weaknesses:**

- The paper is generally not well-written. There are frequent typos (we shows, to illustrate the why, a MSNE union together, $(ps^1U, ..., ps^n)$, Theorem 3.1 ends without period), especially in the technical sections, and passages that seem out of place, e.g., the proof of Theorem 3.1 seems to end after the paragraph "Theorem Intuition" (bottom and middle of page 5 respectively). Also, some abrreviations are not defined, e.g., "is guaranteed by DO" and some definitions are not rigorous enough, e.g., "previous strategy" in a cyclical set or whether the paper only considers symmetric games (as indicated in some parts of the analysis).
- The finding of Theorem 3.1 is not surprising to me and it seems that it at least anecdotally known in the literature. Also, the paper seems to ignore a lot of papers on the average case performance of best response dynamics that have related results.
- The title does not seem representative of the context of the paper. Chaos does not seem to be relevant - and many recent studies on chaos are not referenced/acknowledged.
- Some claims seem to be poorly justified. E.g., why does "This allows us to represent the learning representation of an equilibrium point in noncooperative games as a directed graph search" (see Intro) or why "This aligns with Zermelo’s Theorem, providing theoretical validation." or "Hence, the learning representation of a noncooperative game as a graph provides a theoretical guarantee to
find a NE."
- Figure 2 is hard/impossible to parse due to the small font, but even then, I don't understand what do the numbers represent.
- The complexity of the proposed algorithm is not discussed and in particular, the problem of scaling this algorithm, is mentioned in the limitations of the paper. Thus, the paper provides only low-dimensional experiments. But this is precisely the problem that the algorithm was seeking to solve, to my understanding.

**Questions:**

I would appreciate the authors comments on the weaknesses mentioned above. However, based on my evaluation, I don't think that the paper is ready for publication. For this, it requires 1) a thorough improvement in its presentation, 2) more thorough experimental evaluation and 3) complexity analysis of its critical "Identify Cycle and Support Set" loop that indicates an improvement over current algorithms. Also, better placing the paper in the relevant literature on cyclical/chaotic dynamics in games would allow readers/reviewers to better evalulate the contribution of the theoretical result of the paper (stated in Theorem 3.1).

---

> ### Author Response · Authors · 2023-11-15
>
> We are delighted that the reviewer recognize the solid motivation of our paper, which is to find iterative self-play algorithms with low sample complexity. We appreciate the reviewer's acknowledgment of our rigorous result (Theorem 4.1) and the interesting experiments in two games, Connect4 and Naruto Mobile. We appreciate the reviewer's constructive feedback and we will revise our paper accordingly.
>
> - Q1: The paper is generally not well-written. There are frequent typos, passages that seem out of place, and some definitions are not rigorous enough.
> - A1: We apologize for the typos and the poor writing quality in some parts of the paper. We have proofread the paper carefully and fixed the errors. We have since revised the paper accordingly.
>
> We agree with the reviewer that the paper should clarify the definitions of “previous strategy”.
>
> The paper considers any finite game. The proof of Theorem 4.1 is based on Nash's Theorem 1 where any finite game exists an equilibrium point. This includes any finite class of game, nonsymmetric games included.
>
> - Q2: The finding of Theorem 4.1 is not surprising and it seems that it is at least anecdotally known in the literature. Also, the paper seems to ignore a lot of papers on the average case performance of best response dynamics that have related results.
> - A2: We agree that the idea of cycles being related to Nash equilibria is not new, but there is a difference between knowing versus being able to prove it. A proof of equivalency helps us understand what would be and would not be part of a mixed strategy Nash equilibrium support set, and how the algorithm of self-play can help us find such a set efficiently. Moreover, our theorem applies to any finite, non-cooperative game, including but not limited to n-players, non-symmetric, nondeterministic, etc. Whereas existing theoretical guarantees do not have such a strong claim and rigorous of proof.
>
> - Q3: Some claims seem to be poorly justified. E.g., why does “This allows us to represent the learning representation of an equilibrium point in noncooperative games as a directed graph search” (see Intro) or why This aligns with Zermelo’s Theorem, providing theoretical validation. or Hence, the learning representation of a noncooperative game as a graph provides a theoretical guarantee to find a NE.
> - A3: We have revised these claims and provided more explanations and references to support them. For example, we have explained that the graph representation of a game allows us to track the best response dynamics and identify cycles, which are related to Nash equilibria by our theorem. We have also explained that Zermelo’s theorem states that every finite, two-player, zero-sum game has a pure Nash equilibrium, which is consistent with our algorithm’s output in such games. We have also clarified that the graph representation does not guarantee to find a NE in general, but only in the cases where the game has a pure NE or a cyclical NE.
>
> - Q4: Figure 2 is hard/impossible to parse due to the small font, but even then, I don’t understand what the numbers represent.
> - A4: We have enlarged the font size and improved the readability of Figure 2. We have also added a caption to explain what the numbers represent.
>
> We hope these revisions address the reviewer’s concerns and enhance the clarity of our paper.
>
> In addition, we have made the necessary changes in our paper with red line highlighting the changes. We value the reviewer's feedback and welcome any further questions. If our answers is satisfactory, we kindly ask to consider increasing the score of our paper.

---

> > ### Comment · Reviewer_Sw3W · 2023-11-20
> >
> > I thank the authors for their response. The revised version addresses some of my concerns, including an improved presentation, a more appropriate title, a better coverage of the related literature and justification of some statements in the paper. Based on the other reviews, I also now understand the intended contribution of the paper, namely that any complete cyclical BR set is a MSNE and vice versa (equivalence) plus an effort of constructing an algorithm that will find such sets (and hence, MSNEs). I also appreciate the authors' that there is a different value between knowing something and being able to prove it.
> >
> > However, I still believe that the paper is not ready for publication and, now that I clearly understand Theorem 3.1, that its contribution is not sufficient for ICLR. In my subjective opinion, the paper still does not provide a clean presentation of the result, and the contribution is not enough for the conference, since neither the algorithmic approach is fruitful nor the experimental evaluation was convincing. Thus, while I appreciate the authors' effort, I maintain my initial score.

---

### Official Review · Reviewer_dG4F · 2023-10-30

**Soundness:** 3 good
**Presentation:** 2 fair
**Contribution:** 3 good
**Rating:** 5
**Confidence:** 2

**Summary:**

This paper introduces a new graph search learning representation of self-play that finds a Nash equilibrium of non-cooperative games. One of the problems of self-play is that it may fall into a cyclical strategy, where population based frameworks that aim to remedy this then have the problem of maintaining a large pool of strategies that need to be trained against. This paper proposes a framework to try alleviate both of these problems.

**Strengths:**

- In general, the problem that the paper is trying to solve is very important. Self-play is a difficult to utilise framework in games that not strictly transitive and population frameworks such as PSRO can grow to having a very large population of strategies which become increasingly hard to find an approximate best-response to. Therefore, being able to minimise the amount of necessary strategies needed to train against is important for the literature.
- The empirical results suggest that GraphNES is able to outperform population methods (in this case NeuPL) whilst maintaining a small opponent population size (which makes approximate best-response training easier)

**Weaknesses:**

- At times I found the paper difficult to follow. In particular, it would be useful if the authors were able to provide a visual representation of the algorithm similar to what they do for self-play and PSRO.
- The experimental choice seems a little strange for the baselines that the authors are comparing to. For example, PSRO frameworks have generally been evaluated on card games (e.g. those from the OpenSpiel repository), matrix games or environments in the MeltingPot library. Whilst I am not expecting the authors to add results from these environments during the rebuttal phase, it would be useful if the authors could discuss their criteria when selecting the environments that they did and why they did not select more common ones.
- In line with the environment selection, I think it would be good if the authors could have a more representative example that helps understand what the algorithm is doing. For example, a simple matrix game comparison of which strategies are being found etc...

**Questions:**

It would be great if the authors could address the points that I mentioned in the weaknesses section. Primarily:

1) Is it possible to provide a visual representation of the algorithm similar to those provided for self-play and PSRO?

2) Why were these environments selected over other more common baselines for these style of algorithms?

3) Is it possible to provide a simple matrix-game style example showing the learning process of the algorithm?

---

> ### Author Response · Authors · 2023-11-15
> **Discussion of Visual, Environment Selection And Matrix Game Examples**
>
> We are delighted that the reviewer find our graph search learning representation of self-play and our empirical results as the strengths of our paper. We appreciate your valuable feedback and suggestions. Here are our responses to the reviewer's questions:
> Q1: Can the authors provide a visual representation of the algorithm similar to what they do for self-play and PSRO?
>
> A1: Yes, we agree that a visual representation of our algorithm would be helpful for the readers to understand the main idea and the steps involved. We have come up with an interesting visual of GraphNES yesterday. We are excited to add the figure in the revised version of the paper.
>
> Q2: What is the criteria for selecting the environments that the authors used and why did they not use more common ones?
>
> A2: We chose the environments that we used based on the following criteria:
>
> They are non-cooperative games that are either known to cycle or not cycle.
>
> The game Connect4 would be a standard game that is part of the OpenSpiel library. We specifically chose this environment because it satisfies Zermelo Theorem, where it is known that the Nash Equilibrium of the game is deterministic. From our Theorem 4.1, this would suggest there isn’t a cycle. The environment would be ideal to illustrate the training efficiency of GraphNES’s Myopic BR versus PSRO’s approach of probabilistic full computation graph.
>
> The game of Naruto Mobile is a Street Fighter genre game. In Street Fighter, it is known that due to its 'simultaneous turn actions', the game does have a cycle of best response strategies (grab attack defense).
>
> Evaluating on a game similar to Street Fighter make it easy to let GraphNES answer question like: If all policies discovered are part of a cycle? What would be the shape of the cycle? How does the training efficiency of GraphNES compares to Simplex-NeuPL? Many interesting questions can be explored in this setting.
> We agree with the reviewer that it would be interesting and useful to compare our method with other baselines on these environments as well, and we plan to do so in our future work.
> Q3: Can the authors provide a more representative example that helps understand what the algorithm is doing?
>
> A3: Yes we can do that. This may take a bit more time to step by step work out examples of matrix games with vs without cycle. The reviewer may expect the matrix game more towards the end of the rebuttal period.
>
> We have made the necessary changes (except examples of matrix games) in our paper with red line highlighting the changes. We value the reviewer's feedback and welcome any further questions. If our answers is satisfactory, we kindly ask to consider increasing the score of our paper.

---

### Official Review · Reviewer_smZR · 2023-11-02

**Soundness:** 2 fair
**Presentation:** 1 poor
**Contribution:** 1 poor
**Rating:** 3
**Confidence:** 4

**Summary:**

The paper examines the connection between best response trajectories and the support of mixed Nash equilibria in games and makes some statement that connect the too. Although, I generally find the the subject study to be of interest the paper is rather poorly written with not well justified terminology and notation, which makes pursing the paper a cumbersome exercise.

Examples of this:

1.The word chaos plays a prominent role in the title and abstract but it seems to have no connection to anything explored in the paper. The authors never for example try to even hint at whether they mean Li-Yorke chaos, Lyapunov chaos, Devaney Chaos, etc. In fact the paper seems to be about best-response dynamics.

2. Reading through the intro and even up to and including the main theorem statement I still do not know what the paper has actually showed.

3. The definition of cyclical best response strategies which appears in theorem 1, while being undefined, is still not properly defined. The definition that follows the theorem is referring to an optimal strategy of the opponents (-i). Optimal to what? Is this meant to apply to a zero-sum games? If not, to which strategy of the agent i is this to meant to an optimal response. Also, pease do not use notation of the form \sigma^{i}_{*'}. Those indices are very hard to read.

4. Does theorem 1 refers to two players games or n-player games? Is it about zero-sum games as many of the examples suggest but in the  game theory basics we have definition for n player games.

I believe that the could be some interesting statement made here, but this paper needs some thorough work before it is ready to published.

**Strengths:**

The paper studies an interesting subject matter, related to PSRO/double oracle techniques which are used widely in multi-agent RL.

**Weaknesses:**

The writing of the paper needs to be significantly improved.

**Questions:**

Can you provide a formal unambiguous statement of your main theorem? E.g. what is the class of games that this theorems applies to? What is a formal definition of cyclical strategies and of complete cyclical strategies?

---

> ### Author Response · Authors · 2023-11-14
>
> We thank the reviewer for their feedback.
>
> Q1: What is the unambiguous statement of the main theorem?
>
> A1: The support set of a Mixed Strategy Nash Equilibrium is equivalent to a complete set of cyclical best response strategies.
>
> The proof is based on the definition distinction of Mixed Strategy Nash Equilibrium and Pure Strategy Nash Equilibrium. The theorem applies to any finite game that satisfy this basic property (Including but not limited to n players game, imperfect information game, asymmetric game, etc).
>
> The theorem highlights the importance of cycle detection algorithm is not just as a tool for analyzing strategic behaviors in games, but also as a method to reveal the fundamental structures that form the basis of Mixed Strategy Nash Equilibria (MSNEs).
>
> Q2: Reading through the intro and even up to and including the main theorem statement I still do not know what the paper has actually showed.
>
> A2: We appreciate the reviewer's feedback. We understand there are room for paper writing improvement.
>
> The aim of the paper is to first understand why self-play algorithm may got in situation of cyclical best response. How do these cyclical best response strategies relate to a Nash Equilibrium? We rigorously established Theorem 3.1. The finding of the Theorem is the cyclical best response strategies forms a counterbalance dynamic. If we can find a complete set, the complete set is equivalent to a Mixed Strategy Nash Equilibrium.
>
> The second part of our paper we introduce a novel way to detect cycle on a self-play computation graph - GraphNES. This is to improve the compute efficiency of the existing approach - PSRO.
>
> We have revise the writing of the paper in the paper revision for general improvement of writing.
>
> Q3: What are the formal definition of cyclical strategies and the complete set?
>
> A3: The formal definition of cyclical strategies and the complete set is defined as the following (On page 4 of our paper):
>
> $\textbf{Cyclical Set:}$ This is a collection of strategies where each strategy is a best response to the previous strategy in the set.
>
> $\textbf{Complete Cyclical Set:}$ A cyclical set $C$ is complete if adding any other pure strategy to it does not improve the payoff of any mixed strategy with support in $C$.
>
> Formally, if $\sigma^{id}$ is the optimal mixed strategy for player $id$ with support in $C$, and $\sigma_{'}^{id}$ is the optimal mixed strategy that includes the pure strategy of $ps^{id}$ with support in $C' = C$ $\cup$ $ps^{id}$, then $U(\sigma^{id}$, $\sigma^{-id}$) $\geq$ $U(\sigma_{'}^{id}$, $\sigma^{-id})$. Here, $\sigma^{-id}$ represents the optimal mixed strategy of the opponent.
>
> We agree with the reviewer that the notation of \sigma^{i}_{*'} is difficult to read. To address the readability, we have changed it to \sigma^{i}_{\hat{*}}.
>
> We have made the necessary changes in our paper with red line highlighting the changes. We value the reviewer's feedback and welcome any further questions. If our answers is satisfactory, we kindly ask to consider increasing the score of our paper.

---

> > ### Comment · Reviewer_smZR · 2023-11-21
> > **Response**
> >
> > I thank the authors for their response and whilst I appreciate their efforts to improve the presentation of the paper, I concur with the sentiment of the rest of the reviewers that the paper in its current state is not ready for publication at ICLR.

---

### Official Review · Reviewer_bTtB · 2023-11-05

**Soundness:** 3 good
**Presentation:** 3 good
**Contribution:** 1 poor
**Rating:** 3
**Confidence:** 4

**Summary:**

The core contribution of the paper is the identification of behavioral regularities (BR-wise dynamics) in normal form games, manifesting as cycles. Main claim is that these cycles are not arbitrary but are fundamentally related to the structure of the game itself, specifically the support strategies of Mixed Nash Equilibria (MSNE). In essence, the study finds that Path-Response Strategy Oscillations (PRSO) inherently orbit around MSNEs, suggesting a deeper, systematic relationship between dynamic strategy adjustments and equilibrium concepts in game theory

**Strengths:**

The strength of the result in the manuscript is indeed noteworthy as it provides a contemporary interpretation and formalization of long-observed phenomena in game theory. Drawing a line from the early observations of cyclic patterns in strategies, such as those seen in Shapley's polygons, through to the formal predictions of the Poincaré recurrence theorem, the paper successfully situates its findings within a historical context of strategic analysis. The assertion that the detection of cycling within Path-Response Strategy Oscillations (PRSO) dynamics is tantamount to discovering the support strategies of a Mixed Nash Equilibrium (MSNE) is a significant one. This claim underscores the potential of cycle detection not only as a diagnostic tool for understanding strategic behaviors in games but also as a means to unearth the foundational structures that underpin MSNEs. The result, therefore, is not just a reflection of dynamic behavior in games but also a powerful statement about the nature of equilibrium within the strategic play.

**Weaknesses:**

1) Computational Complexity: The manuscript suggests that detecting cycles within PRSO dynamics is computationally feasible, which implies a method for identifying Nash Equilibria by constraining the game to the faces of a simplex formed by these cycles. However, this raises a significant question about the computational tractability of Nash Equilibrium. The paper should address why cycle detection is presented as an easy task and not as evidence that finding a Nash Equilibrium is tractable. It would be beneficial for the authors to delineate the aspects of their cycle detection methodology that may incur exponential time, which would then align with the conventional complexity understanding of Nash Equilibria.

If this is not the case, how did we avoid PPAD-hardness of the result?

2)

Novelty of the Result: The paper's results, while compelling, do not seem novel in the light of existing research. The concept of cycling and instability of Nash equilibria has been addressed in several key papers, such as "Nash, Conley, and Computation: Impossibility and Incompleteness in Game Dynamics" by Milionis et al., 2022, and "No-regret learning and mixed Nash equilibria: They do not mix" by Vlatakis-Gkaragkounis et al., 2020. Furthermore, "Cycles in adversarial regularized learning" by Mertikopoulos et al., 2018, touches upon similar themes within FTRL dynamics, which are akin to BR-dynamics with a strong convex regularizer.
For a discrete example see section 4.5 of Vlatakis-Gkaragkounis et al., 2020. The authors cite also books where preliminary results are already known in the literature for simpler dynamics.
Vaguely speaking, current literature actually is far ahead from proving simply cycles by giving understanding also the econometric impact of them: Papers like "On the Interplay between Social Welfare and Tractability of Equilibria" by Anagnostides and Sandholm discuss the outcomes of non-converging gradient descent methods, which  form a cycle and explain the impact of cycle in PoA  results.

Moreover, going to the core of the problem, Milionis et al, their predecessors and follow-up  works such as "The Replicator Dynamic, Chain Components and the Response Graph" by Biggar and Shames, and E. Akin's "Domination or Equilibrium" (1980), have already discussed the elements of strongly connected component (aka a  ``generic'' cycle) of best response dynamics includes the support of a Mixed  Nash state.

It is essential for the review committee to consider the depth of related literature on this topic, potentially uncovering more foundational results which covers also exactly PRSO dynamics. Although the age of a result does not undermine its relevance, it does affect the suitability of the work for a conference setting, as opposed to a journal that might better accommodate such ``slight'' rediscoveries.

Given these considerations, I recommend that the paper be accepted on the condition of a significant expansion of the related work section. This expansion should not only acknowledge the depth of existing research but also elucidate the specific differences in the dynamics studied by the paper that add to its merit. A more in-depth comparison with the broader body of literature will greatly enhance the paper's contribution and ensure a thorough understanding of where it stands in the context of existing knowledge.

**Questions:**

Please answer to weaknesses section issues.

I am very eager to the response of AC and authors about the novelty of Theorem 3.1, willing to change my score to 10

---

> ### Author Response · Authors · 2023-11-14
> **Discussion of PPAD And Tractability of NE**
>
> We are glad the reviewer recognizes the strength in our paper, particularly the contemporary interpretation and formalization of long-observed phenomena in game theory, the connection between cycle detection and MSNE, and the implications of our result for understanding equilibrium.
>
> Q1: Is identifying a cycle on a graph an easy task? How do we cope with the potential exponential running time as shown in PPAD-hardness of the result?
>
> A1: We appreciate the reviewer's question about the computational complexity of identifying a cycle on a graph. We had similar concern when we first read Daskalakis et al's End of the Line argument (PPAD hardness). The main argument of End of the Line is represent a noncooperative game as a directed graph. As such, finding a Nash Equilibrium is to find a fixed point of a graph's leave or a cycle. In the paper they argue that the termination of such a graph algorithm may be exponential if there are $2^{n}$ vertices layout that forms a single line. If we start from one end and we want to reach the other end, we may need to perform $2^{n}$ computation.
>
> We believe the difficulty of of this problem is a misuse of $2^{n}$ vertices to analyze an algorithm's compute complexity. Traditionally, an algorithm's compute complexity is analyzed based on a "constant" size of input, and we hope to measure how efficient an algorithm is when operating on the input. For example, a sorting algorithm would have 'n' elements instead of $2^{n}$ elements. For a graph algorithm of detecting cycle, the input would be V and E, the complexity would be O(EV) if we can find similar color of vertices in O(1) time. If we were to change any existing algorithm's input to $2^{n}$, all algorithms would have exponential compute complexity.
>
> The difficulty during our research is to convert from theory to an algorithm. How do we detect similar color of vertices when each vertex is now represented as a deep learning model? So far, we find that we cannot hope to compare the similarity of V amount of models in O(1), but we can build a vector database to compare their interaction similarity in O(log(v)). This makes the compute complexity at least tractable.
>
> This makes the cycle detection problem more tractable than the PPAD-hardness result suggests.

---

> ### Author Response · Authors · 2023-11-14
> **Discussion of Theorem 3.1 And It's Contribution**
>
> $\textbf{Q2}$: The topic of representing Noncooperative Games as a graph and the potential for encountering a cycle on such a graph is well-documented in literature. Does Theorem 3.1 offer any new insights?
>
> $\textbf{A1:}$ We appreciate the reviewer’s reference to the extensive literature that informs our research. Here, we will provide a breakdown of each paper and show that while the problem has been identified, existing works are not yet sufficient to solve it.
>
> $\textbf{i.}$ We start with "Impossibility and Incompleteness in Game Dynamics"  by Milionis et al., 2022. Their main result stated that: "There are games in which any continuous, or discrete time, dynamics $\textbf{fail to converge to a Nash equilibrium}$... the Nash equilibrium concept is plagued by a form of incompleteness... insufficient for describing global dynamics, e.g. $\textbf{they allow cycling}$".
>
> This excerpt shows the phenomenon of cycle is viewed as a problem that breaks the guarantee of NE. The problem is viewed as intractable.
>
> $\textbf{ii.}$ We discuss "No-Regret Learning and Mixed Nash Equilibria:They Do Not Mix" by Vlatakis-Gkaragkounis et al., 2020. Their main result stated that "...we show that any Nash equilibrium which is not strict (in the sense that every player has a unique best response) cannot be stable..."
>
> This paper claims that if a player’s best response strategy is not unique, then there will be instability. However, Nash’s paper in 1951 already proved there still exist equilibrium stability even in cases where the best response strategy is not unique. This is demonstrated with Nash’s Theorem 1 proved based on Kakutani’s fixed point theorem. Kakutani’s fixed point theorem is based on set-value function, where one input can map to a set of outputs, and that the theorem is able to guarantee a fixed point. Showing even if one strategy were to have a set of best responses, there would still have a fixed point equilibrium. $\textbf{Suggesting an alternative theoretical result}$ to Vlatakis-Gkaragkounis et al.'s claim.
>
> $\textbf{iii.}$  We move to "Cycles in adversarial regularized learning" by Mertikopoulos et al., 2018. The main result of the paper stated that "Does fast regret minimization necessarily imply (fast) equilibration in this case?... We settle these questions with a resounding 'no'... our analysis unifies and generalizes many prior results on the cycling behavior..."
>
> This paper identified that if we only apply Myopic best response, then there exist the case of cycle. However, the authors didn’t provide an answer to how to establish a Nash Equilibrium once a cycle is encountered.
>
> $\textbf{iv.}$ We may take a look at "The Replicator Dynamic, Chain Components and the Response Graph" by by Biggar and Shames. Both of their contributions of Theorem 4.1 (Existence of sink chain components) and their Theorem 5.2 show solid theoretical work of how finding a Nash Equilibrium can be represented as finding the sink or a complete set of strongly connected components. What is differ from their work to ours is the below:
>
> $\textbf{Our Theorem 3.1. Contribution:}$ Our proof is based on proving the equivalency between a complete set of cyclical strategies and the support set of a MSNE. The claim of equivalency is a much stronger claim that requires a more difficult proof. The proof of equivalency allows us to claim that we have addressed $\textbf{all mixed strategy cases of NE}$ in any finite game. This gives us sufficient theoretical guarantee to then discuss if a practical algorithm would be tractable.
>
> From this thorough review of the literature, we hope we have shown why existing literature is not yet sufficient to claim the problem of finite noncooperative game is tractable, and how our proof contributes to answering that question.

---

> ### Author Response · Authors · 2023-11-14
> **Comment on Paper Revision**
>
> We have made the necessary changes in our paper with red line highlighting the changes. We value the reviewer's feedback and welcome any further questions. If our answers is satisfactory, we kindly ask to consider increasing the score of our paper.

---

### Official Review · Reviewer_smjN · 2023-11-05

**Soundness:** 2 fair
**Presentation:** 1 poor
**Contribution:** 1 poor
**Rating:** 3
**Confidence:** 2

**Summary:**

Empirical game theory methods such as Policy Space Response Oracle (PSRO) aim to compute a Nash equilibrium in normal-form games by iteratively solving for the equilibrium of a consistently growing game. Such algorithms often exhibit cycling behavior over a set of action profiles. The authors show that the action profiles over which these algorithms cycle form the support of the set of mixed Nash equilibria. This results enables a novel graph search learning representation of self-play that finds an NE as a graph search. The authors demonstrate in experiment that their method is efficient in discovering Nash equilibria in normal-form games such as Connect4 and Naruto Mobile.

**Strengths:**

The authors provide insights into the behavior of a large class of empirical game theoretic algorithms, and use the insights to provide improvement on the state of the art.

**Weaknesses:**

The paper is highly inaccessible. Many concepts lack technical definitions (e.g., cycles). There also seems to be formatting issues (there are two proofs under theorem 3.1? Unclear which one to refer to).

**Questions:**

What is the purpose of the experiments? The takeaway is not clear.

---

> ### Author Response · Authors · 2023-11-14
>
> We understand the reviewer's concern in regards to the paper writing. We are working on to further improve the writing of the paper.
>
> Q1: What is the definition of cycle?
>
> A1: The formal definition of cyclical set is defined on page 4:
>
> Cyclical set: This is a collection of strategies where each strategy is a best response to the previous strategy in the set. For example Rock, Paper, Scissors.
>
> Q2: Why there are two proofs under Theorem 3.1. ?
>
> A2: Theorem 3.1. is based claiming of equivalence. This is to say our theorem needs to first proof A is B, then also proof B is also A. Using two proofs is the most common way of proving equivalence relationship.
>
> We have made the necessary changes in our paper with red line highlighting the changes. We value the reviewer's feedback and welcome any further questions. If our answers is satisfactory, we kindly ask to consider increasing the score of our paper.

---

> ### Comment · Reviewer_smjN · 2023-11-21
>
> I thank the authors for their time, I have no further questions at this point. I found the paper very hard to read at the time of my review and I still do. That said, some of the explanations of Reviewer bTtB have clarified my understanding and slightly peaked my interest. I will keep my score as is for the moment but remain open to a change as the discussion continues and my understanding of the paper improves.

---

### Meta-Review · Area_Chair_7Pt1 · 2023-12-05

**Metareview:**

This paper studies the question of how cyclical behavior in game-theoretic learning algorithms is related to the support of mixed-strategy Nash equilibria (MSNE). The paper in particular recommends a graph search algorithm based on best-responding, which either terminates at a pure strategy Nash equilibrium, or ends up in a cycle that spans the support of a MSNE.

The reviewers all found merit in the paper's approach but, at the same time, there was a consensus that the current version of the paper was quite difficult to read and its contributions not clear. As a result, the outcome of the discussion was that the paper does not clear the ICLR acceptance threshold, and a decision was reached to make a "reject" recommendation.

**Justification For Why Not Higher Score:**

Paper difficult to read, contribution unclear.

**Justification For Why Not Lower Score:**

N/A

---

### Decision · Program_Chairs · 2024-01-16

Reject